# Optimal Position and Distribution Mode for On-Site Hydrogen Electrolyzers in Onshore Wind Farms for minimal LCoH

Thorsten Reichartz[1], Georg Jacobs[1], Tom Rathmes[1], Lucas Blickwedel[1], Ralf Schelenz[1]

[1]Chair for Wind Power Drives, RWTH-Aachen University, Aachen, 52074, Germany

*Correspondence to*: Thorsten Reichartz (thorsten.reichartz@cwd.rwth-aachen.de)

**Abstract.**

Storing energy is a major challenge in achieving a 100 % renewable energy system. One promising approach is the production of green hydrogen from wind power. This work proposes a method for optimizing the design of wind-hydrogen systems for existing onshore wind farms in order to achieve the lowest possible Levelized Cost of Hydrogen (LCoH). This is done by

application of a novel Python-based optimization model, that iteratively determines the optimal electrolyzer position and distribution mode of Hydrogen for given wind farm layouts. The model includes the costs of all required infrastructure components. It considers peripheral factors such as existing and new roads, necessary power cables and pipelines, wage and fuel costs for truck transportation and the distance to the Point of Demand (POD). Based on the results, a decision can be made whether to distribute the hydrogen to the POD by truck or pipeline.

For a 23.4 MW onshore wind farm in Germany, minimal LCoH of 4.58 € $kg_{H2}^{-1}$ at an annual hydrogen production of 241.4 $t_{H2}\ a^{-1}$ are computed. These results are significantly affected by the position of the electrolyzer, the distribution mode, varying wind farm and electrolyzer sizes, as well as distance to POD. The influence of the ratio of electrolyzer power to wind farm power is also investigated. The ideal ratio between rated power of electrolyzer and wind farm lies at around 10 % and a resulting capacity factor of 78 % for the given case.

The new model can be used by system planners and researchers to improve and accelerate the planning process for wind-hydrogen systems. Additionally, the economic efficiency, hence competitiveness of wind-hydrogen systems is increased, which contributes to an urgently needed accelerated expansion of electrolyzers. The results of the influencing parameters on LCoH will help to set development goals and indicate a path towards cost-competitive green wind-hydrogen.

## 1 Introduction

The International Energy Agency (IEA) projects that global demand for hydrogen will nearly double by 2030 from 2021 levels. Today, less than 1 % of the world's hdydrogen production is low-emission hydrogen, while 99 % is produced either from fossil fuels or as a by-product (IEA, 2021). To meet the future demand for green hydrogen, the European Union has set a target of 40 GW of installed electrolyzer capacity in 2030 (European Commission, 2020). Part of the electrolysis capacity will be built in combination with wind farms, as encouraged by the European Commission (European Commission, 2023). In this way,

electrolyzers can reduce grid-induced curtailment of wind turbines, increase the utilization of wind farms, and enable the storage of large amounts of renewable energy in the form of hydrogen. However, the rate of electrolyzer deployment is currently low at less than $0.5\ GW\ a^{-1}$ in the EU (Ueckerdt et al., 2021). There are a number of reasons for this, but one of the most important is the high Levelized Cost of Hydrogen (LCoH). Main drivers for the high LCoH of green hydrogen are the high investment costs for electrolyzers and the electricity costs (Ajanovic et al., 2022).

One possibility to reduce LCoH is to further reduce the Levelized Cost of Electricity (LCoE) of wind turbines and farms. However, the LCoE of wind energy are already low and are unlikely to fall by orders of magnitude in the near future (Bošnjaković et al., 2022). As a result, energy costs, which account for about 40 % of LCoH, will remain high (Ajanovic et al., 2022). The capital cost of electrolyzers will decrease in the future, due to scale up effects and further technology improvements (IRENA, 2020). But wind farm planning has little to no influence on these developments. Therefore, in order

to reduce the LCoH, wind farm developers will need to take advantage of the freedom in the design of the wind-hydrogen system. Numerous studies have addressed the subject, including Hofrichter et al. (2023b), who investigated the optimal power ratio of electrolyzers and renewable energy sources. Their analysis covered wind farm sites characterized by varying full load hours (FLH), but did not consider hydrogen transportation costs nor on-site electrolyzer positioning. Similarly, Schnuelle et al. (2020) and Benalcazar and Komorowska (2022) take the macroscopic approach of evaluating sites based on FLH, neglecting

hydrogen transport and microscopic assessments that include ancillary infrastructure requirements such as existing roads and water pipelines. In their study on hydrogen production from floating offshore wind, Ibrahim et al. (2022) adress the transportation of energy to shore in the form of hydrogen or electricity, considering the central role of energy distribution within energy systems. The study focuses on offshore wind to hydrogen, which limits its transferability to onshore farms. Sens et al. (2022) investigate the ideal locations on a continental and regional scale for hydrogen production from wind and solar to

provide hydrogen to Germany, including hydrogen transportation costs, but they only consider pipeline transportation as they focus on large quantities of produced hydrogen. The authors also made it explicit that they excluded transmission costs for electricity and water on-site. While other studies have also analyzed the costs of the necessary infrastructure for hydrogen production and transportation at the macroscopic level (Yang and Ogden, 2007; Reuß, 2019; Correa et al., 2022), transferable models for a specific cost analysis at the wind farm level, including detailed site-specific infrastructure, electrolyzer positioning

and transport mode optimization, are not available. This publication aims to address and fill that gap by answering the following research question and sub questions:

- **To what extent can wind farm operaters and developers reduce the LCoH of green hydrogen produced at wind farm sites?**
  - What are relevant influencing factors on LCoH of on-site wind hydrogen systems?
- How can those be modelled?
  - What level of LCoH can be achieved, and what is the ideal electrolyser/wind farm power ratio to achieve this minimum, taking into account hydrogen transport and all required infrastructure at a specific wind farm site?

Despite the environmental benefits of green hydrogen, its production costs must be reduced in order to compete with grey hydrogen (Ajanovic et al., 2022). Decentralized hydrogen production brings its own challenges, such as the need to position electrolyzers on wind farm sites, establish deionized water and electricity supply, and transport the hydrogen off-site.

To address this issue, this paper introduces a new methodology that can generate wind farm specific preliminary designs of the entire wind-hydrogen system and compute corresponding LCoH. In order to minimize the LCoH, the electrolyzer position and rated power are optimized considering site-specific conditions such as wind farm power and wind conditions. In addition, the hydrogen distribution mode to the Point of Demand (POD) is optimized. The developed method is based eniterly on open source software.

In Sect. 2, the underlying physical and economic assumptions for dimensioning and selecting the system components are described. In addition, the objective function and the developed optimization algorithm are introduced. In Sect. 3, the results for a case-study wind farm in Germany are presented. Sect. 4 discusses the results and model limitations, and provides an outlook for further research and application.

## 2 Methodology

The method described in the following allows the preliminary design of a cost-optimal on-site wind-hydrogen system for onshore wind farms. Optimal for this study means that minimal LCoH are achieved, while all boundary conditions are met. The developed method can be applied to all onshore wind farm sites, although financial parameters need to be adapted regionally.

The combination of electrolyzers with wind farms comes with a large number of degrees of freedom in design. Making simplifying assumptions is imperative in order to manage complexity, ensure transferability and keep the required computing power within feasible limits.

In Sect. 2.1 and 2.2 all underlying assumptions of the conversion of electricity to hydrogen, storage on site and the hydrogen distribution mode are given. In Sect. 2.3 the applied optimization method and the required input data is explained.

The overall goal of a wind-hydrogen system is to generate hydrogen at a wind farm site and transport it to a POD while minimizing cost. The respective *objective function* is given in Eq. 1. The LCoH are dependent on the Total Expenditures (TOTEX) and the annual mass of hydrogen produced $M_{H2}$.

$$\min(LCoH_{p,d} = \frac{\text{TOTEX}_{p,d}}{M_{H2}})$$

(1)

The calculation of TOTEX is performed using the annuity method, as shown in Eq. 2 and Eq. 3. The weighted average cost of capital ($WACC$) is assumed to be 7 %, as is often used in other studies focusing on renewable energies (Satymov et al., 2022; Fasihi and Breyer, 2020). The costs of hydrogen transportation to the POD are included in the modelling of LCoH. Capital Expenditures (CAPEX) and Operational Expenditures (OPEX) of the system components $i$ depend on the selected hydrogen

distribution mode $d$. A total of seven different possible hydrogen distribution modes are considered. These are derived from

the possible combinations of trailers and diesel or hydrogen trucks and the distribution of hydrogen by pipeline.

Some TOTEX-components are also dependent on the electrolyzer position $p$ at the wind farm site, e.g. power cables, water

pipelines and roads. The lifetime of each component is considered via the parameter $n$, given in years $a$.

$$TOTEX_{p,d} = \sum_{i}^{j}(CAPEX_{p,d,i} \cdot crf + OPEX_{p,d,i}), \tag{2}$$

$$crf = \frac{WACC \cdot (1+WACC)^n}{(1+WACC)^n - 1} \tag{3}$$

Figure 1 shows the boundaries of the wind-hydrogen system. CAPEX and OPEX for all components of the hydrogen system

are included in the calculation of LCoH. However, wind farm costs are not considered. This is based on the assumption that

the wind farm already has a power grid connection and that its layout is unchanged during the hydrogen layout optimization

process. The method currently focuses on the optimization of wind-hydrogen systems for already existing wind farms. The

LCoE and the generation profile of the wind farm serve as input variables. Costs for infrastructure at the POD are not included.

This does not apply for components necessary for unloading the hydrogen trailers and converting hydrogen back into a gaseous

state. This ensures LCoH comparability between different distribution modes.

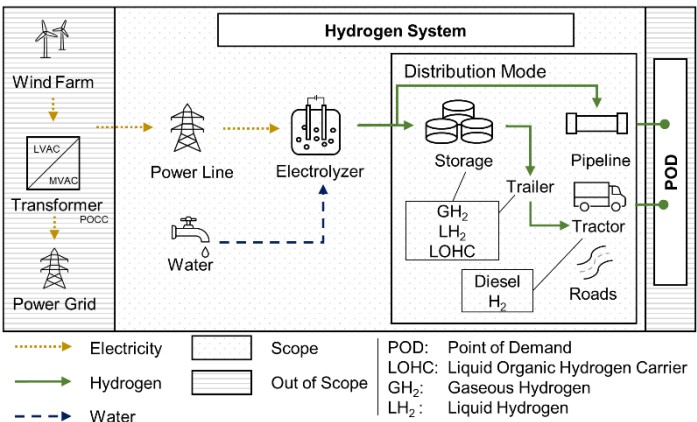

**Figure 1: Overview of the wind-hydrogen system and system boundaries. Excluding illustration of supporting components.**

## 2.1 Hydrogen production

Electrolyzers utilize electricity to split water into hydrogen and oxygen. In this model, the required electricity for the

electrolysis process is acquired solely from the wind farm. No additional electricity is purchased from the grid to feed the

electrolyzer. In this Sect. the electrolyzer and all its auxiliary system components are described. In addition, it is explained

how the utilization of the electrolyzer capacity is approximated depending on the given wind farm and the local wind

conditions.

## 2.1.1 Electrolyzer

Various water electrolysis processes exist, differentiated by the applied electrolyte. The most relevant technologies are the Alkaline Electrolysis, the Proton Exchange Membrane Electrolysis and the Solid Oxide Electrolysis (SOEL). SOEL is still in the development stage and is therefore not included in this study (Buttler and Spliethoff, 2018). There is a trend towards the usage of Proton Exchange Membrane Electrolyzers (PEMEL) for on-site hydrogen production over the use of Alkaline Electrolyzers (AEL). Since PEMEL have better load flexibility, shorter cold and warm start times and allow higher load gradients than AEL, only PEMEL are considered in this study (Buttler and Spliethoff, 2018; Davoudi et al., 2022; Schiebahn et al., 2015; Hermesmann et al., 2021). Estimating the future cost development of electrolyzers is subject to a number of uncertainties, such as the R&D funding and production scale-up effects (Schmidt et al., 2017). Currently, the costs for PEMEL amount in the range of 700 to 1,400€ $kW_{el}^{-1}$ (IRENA, 2020). The specific PEMEL costs for this study are therefore estimated to be 1,000 € $kW_{el}^{-1}$.

In current manufacturer specifications for PEMEL, the efficiencies vary widely, ranging from 52 % up to 69 % (Buttler and Spliethoff, 2018). Given that PEMEL are a relatively new technology with anticipated efficiency improvements in the near future, $\eta_{El}$ is assumed to be 70 % (Reuß, 2019). The efficiency of the electrolysis process also depends on the load at which the PEMEL is operated (Yodwong et al., 2020). However, this correlation is neglected here.

The electrolyzer utilization, here referred to as Capacity Factor $CF_{El}$ must be calculated site specifically in order to be able to calculate the annual hydrogen production $M_{H2,a}$ of a wind-hydrogen system. This is not trivial since wind energy is a volatile energy source. $CF_{El}$ is defined as the percentage of hours per year during which the electrolyzer is operated at equivalent rated power $P_{El}$, as given in Eq. 4. The energy available for the electrolyzer over a full year $W_{El}$, is visualized in Figure 2 and its calculation explained below.

$$CF_{El} = \frac{W_{El}}{P_{El} \cdot 8760\ h}, \forall P_{El} \in (0, P_{Farm}] \tag{4}$$

$W_{El}$ depends on the amount of electricity generated by the connected wind farm. This energy is defined as the Annual Energy Production (AEP). It is assumed that the difference between the AEP and $W_{El}$ is fed to the electricity grid. In practice, the accuracy of the AEP estimation can be enhanced by data availability at the wind farm site, e.g. historical SCADA data.

Estimating the AEP based on the sorted Annual Load Curve (sALC) is possible with minimal available data. The sALC is calculated based on the power curve of the turbines used and the Weibull distribution of wind speeds at rotor hub height at the site (Hau, 2016). It is usually calculated for a single turbine. To obtain the sALC of a wind farm, the curve is multiplied by the number of turbines in the farm. This simplification is assumed to be sufficient for the subject of this work. However, a more accurate sALC, considering wind turbine (WT) positions and wake effects can be generated, for example, using the methodology described by Shapiro et al. (2019) or one of the wake models discussed by Brusca et al. (2018). As shown in Figure 2, based on the sALC and the rated power of the electrolyzer $P_{El}$, the equivalent $FLH_{El}$, $W_{El}$ and thus $CF_{El}$ is computed.

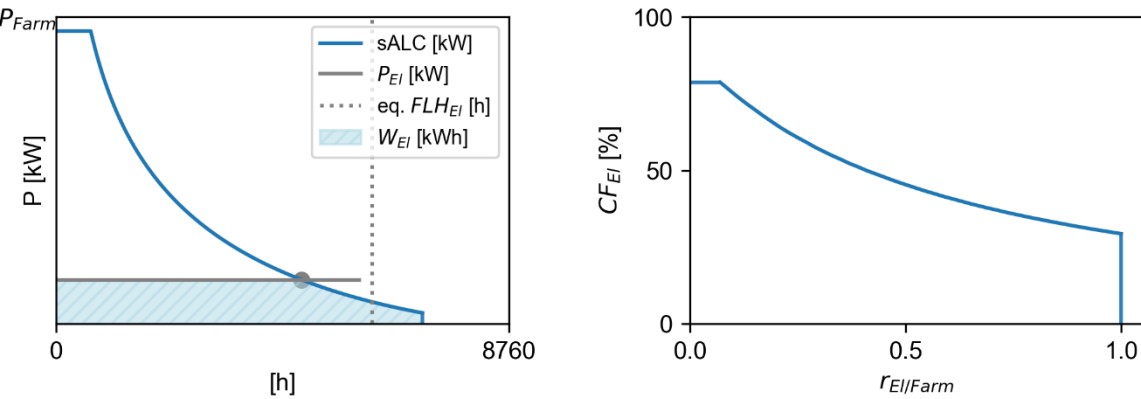

**Figure 2: sorted Annual Load Curve for a wind farm site (left) and $CF_{El}$ depending on the ratio $r_{El/Farm}$ (right)**

In the graph shown on the right in Figure 2, the correlation of $CF_{El}$ with the ratio of $P_{El}$ and $P_{Farm}$ ($r_{El/Farm}$) is visualized. It shows that $CF_{El}$ does not exceed a maximum value of approx. 0.8, as the wind farm does not produce electricity throughout the entire year.

The annual hydrogen production $M_{H2,a}$ is calculated based on the rated electrolyzer power $P_{El}$, the efficiency $\eta_{El}$ and $CF_{El}$, as shown in Eq. 5. The lower heating value of hydrogen $LHV_{H2}$ is 33.33 $kWh\ kg_{H2}^{-1}$ (Adolf et al., 2017).

$$M_{H2,a} = \frac{P_{El} \cdot \eta_{El} \cdot CF_{El} \cdot 8760\ h}{LHV_{H2}} \tag{5}$$

By setting the input parameter $P_{El}$, it is now possible to calculate $M_{H2,a}$ for a specific wind farm and electrolyzer setup.

### 2.1.2 Electrolyzers power connection

PEMEL operate at 1.4 to 2.5 V DC (IRENA, 2020), whereas state of the art WT usually produce 690 V AC, which is transformed to medium voltage (10-35 kV) in the turbine (Žarković et al., 2021). The electricity is then accumulated at the Point of Common Coupling (POCC). In this study, it is assumed that the wind farm remains interconnected with the power grid. Consequently, costs for the transformer required at the POCC to adjust the voltage to grid level are not included in the LCoH calculation.

The power cable of the electrolyzer is connected to the POCC. The cable from the POCC to the electrolyzer is modeled as a 33 kV AC underground cable, as commonly used for electrical wind farm networks (Žarković et al., 2021). Its length $l_{cable}$ is the geodesic length between the position of the POCC $p_{POCC}$, and the position $p_{El}$ of the electrolyzer. $l_{cable}$ includes an additional safety factor $s_{cable}$ which is set as 1.7 to consider terrain and obstacles (Zarkovic et al., 2019).

The electrolyzer power $P_{El}$ affects the required cable thickness and thus the cable costs. Typically, copper or aluminum cables are used, which can be purchased in a wide variety of diameters. Copper cables are used for this study. Based on the cable costs used by Žarković et al. (2021), the specific cable costs are approximated to be 4.56 € $kW_{El}^{-1}\ km^{-1}$ plus installation costs. Installation costs are set to 30,000 € $km^{-1}$ (Hau, 2016). Thus, a linear correlation between the cable costs and $P_{El}$ is assumed. Furthermore, transmission losses are not considered in this work, since cables will mostly cover short distances. An additional

converter transformer is required at the electrolyzer to rectify the current for the PEMEL and reduce the voltage level. Following Fasihi and Breyer (2020), the converter has specific capital costs of $150 \, € \, kW_{El}^{-1}$.

### 2.1.3 Electrolyzers water supply

In addition to electricity, the electrolyzer needs water supply. Stoichiometrically $9 \, kg_{H2O} \, kg_{H2}^{-1}$ is necessary for the electrolysis process (Eq. 6). Including losses and additional 25 % water consumption for equipment cleaning, the real water consumption is approximately $14 \, kg_{H2O} \, kg_{H2}^{-1}$ (Simoes et al., 2021).

$$2 \, H_2O \rightarrow 2 \, H_2 + O_2 \tag{6}$$

Water demand can be provided from various sources, e.g. indstrial wastewater or groundwater. However, additional water treatment is required and not all water sources are available at every location. Therefore, water consumption is modeled using water from the water grid. For wind farm sites in Germany, the water price is set to $2 \, € \, m_{H2O}^{-3}$ (Statistisches Bundesamt, 2020). In the future, globally increasing water scarcity will make an individual consideration of the water supply situation on site imperative.

Despite the good water quality, impurities must be removed from the water by reverse osmosis process, which requires water tanks and pumps at the wind farm site. To avoid detrimental effects on components, PEMEL use deionized water (Guo et al., 2019), whereas the de-ionization process is typically part of the electrolyzer unit, so no additional costs are included here (Simoes et al., 2021). The remaining costs are divided into CAPEX and OPEX and depend on the annual water usage $\dot{V}_{H2O}$, which in turn is dependent on the annual produced amount of hydrogen $M_{H2,a}$ (see Eq. 5). The specific CAPEX are assumed to be $0.6 \, € \, a \, m_{H2O}^{-3}$ and the specific OPEX are assumed to be $0.52 \, € \, a \, m_{H2O}^{-3}$, following Simoes et al. (2021) who conducted a detailed study on water usage of electrolyzers for Portugal. Additionally, the specific water pipeline costs are assumed to be $115 \, € \, m^{-1}$. The required pipeline length $l_{H2O}$ is the geodesic length between the water connection point $p_{H2O}$, and the electrolyzer position $p_{El}$. $l_{H2O}$ including a safety factor $s_{H2O}$ of 1.7 (see Sect. 2.1.2). All additional parameters are given in Table 1.

**Table 1: Financial parameters of hydrogen production and supply infrastructure (Reuß, 2019; Fasihi and Breyer, 2020; Zarkovic et al., 2019; Žarković et al., 2021; Simoes et al., 2021; Statistisches Bundesamt, 2020; Hau, 2016)**

| component | CAPEX | OPEX | lifetime | efficiency |
|---|---|---|---|---|
| electrolyzer | $1{,}000 \, € \, kW_{el}^{-1} \cdot P_{El}$ | $3 \, \% \, a^{-1} \cdot CAPEX$ | 10 a | 70 % |
| power cable | $(4.56 \, kW^{-1} \cdot P_{El} + 30{,}000) \, l_{cable} \, € \, km^{-1}$ | $1 \, \% \, a^{-1} \cdot CAPEX$ | 50 a | 100 % |
| converter transformer | $150 \, € \, kW^{-1} \cdot P_{El}$ | $1 \, \% \, a^{-1} \cdot CAPEX$ | 50 a | 98.6 % |
| water pipeline | $115 \, € \, m^{-1} \cdot l_{H2O} + 0.6 \, € \, a \, m^{-3} \cdot \dot{V}_{H2O}$ | $2 \, € \, m^{-3} + 0.52 \, € \, a \, m^{-3} \cdot \dot{V}_{H2O}$ | 10 a | / |

## 2.2 Hydrogen distribution

Unlike for the transport of electricity, there is no comparable distribution network for hydrogen. Hence, transportation of decentral generated hydrogen is unevenly more complex than distributing electricity. For some hydrogen production sites, pipelines may be viable, while other sites are better served by trucks. An additional degree of freedom is the possible options of transporting hydrogen in trailers in liquid (LH$_2$) or gaseous (GH$_2$) state or bound using liquid oxygen hydrogen carriers (LOHC). In this Section, the necessary assumptions to individually select the most cost-effective distribution mode for a wind farm site are explained. Figure 3 shows an overview of the different hydrogen distribution modes and their impact on the required infrastructure. Although the components for dehydration, as well as vaporizers, are not located at the wind farm, but at the customer's site, their costs are included. This ensures the comparability of the LCoH of all distribution modes, as transport in other forms may require the hydrogen to be reconverted at the costumer's site.

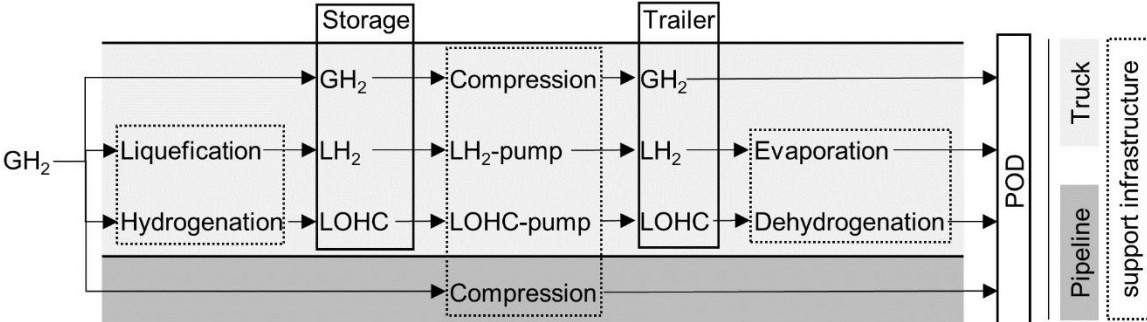

**Figure 3: Overview of considered hydrogen distribution modes and their impact on required infrastructure components**

### 2.2.1 Hydrogen storage

Hydrogen produced at the wind farm site must be temporarily stored before it is transported by truck, resulting in the need for hydrogen storage units. In case of distribution by pipeline, additional storage is not required, since hydrogen is continuously carried off.

There are various approaches to store hydrogen. The use of salt caverns as natural storage is promising for storing large volumes of hydrogen (Caglayan et al., 2020). However, since salt caverns are not available in all locations and the quantities of hydrogen produced are comparatively low, they are not considered further. Therefore, as shown in Figure 3, GH$_2$, LH$_2$ or LOHC storage units are used, as proposed by Reuß (2019). Storage costs are mainly driven by the storage type and its required size. For all storage types, losses are neglected, as they are regularly discharged and storage losses during a day are less than 1 % for all storage types (Reuß, 2019). According to the approach of Yang and Ogden (2007), the storage units used have a capacity $c_{H2,stor}$ of 50 % of the maximum daily hydrogen production $M_{H2,d,max}$, as given in Eq. 7.

$$c_{H2,stor} = 0.5 \cdot M_{H2,d,max} = 0.5 \cdot \frac{P_{El} \cdot \eta_{El} \cdot 24\,h}{LHV_{H2}} \tag{7}$$

All necessary parameters to calculate the TOTEX of each storage type are given in Table 2. The impact of the selected storage type, on the required support infrastructure at the wind farm is shown in Figure 3. All other parameters used to calculate the cost of the support infrastructure are given in Table A1.

In case a LH$_2$ tank is used, a hydrogen liquefier is required. The hydrogen is cooled down below its boiling point and compressed, which requires up to 15 $kWh_{el} \, kg_{H2}^{-1}$ (Reuß, 2019). However, dedicated studies on the liquefication process assume an energy consumption of 6.76 $kWh_{el} \, kg_{H2}^{-1}$, which is used here (Stolzenburg et al., 2013). The investment costs for liquefiers are high and depend on the maximum daily hydrogen production $M_{H2,d,max}$ (see Table A1), which has to be considered when selecting the distribution mode. The advantages of LH$_2$ are lower storage costs and higher density compared to GH$_2$.

In case of using a LOHC tank, hydrogenation of a typically aromatic compound is used, which requires an energy input of 9.08 $kWh_{th} \, kg_{H2}^{-1}$, which depends on the compound used. The necessary thermal energy is provided by the conversion of electricity supplied by the wind farm. Recovery and further usage of thermal waste energy, is not considered, although 8.8 $kWh_{th} \, kg_{H2}^{-1}$ are emitted during the process (Müller et al., 2015). The costs for the hydrogenation unit are estimated based on Reuß (2019) and depend on the maximum daily hydrogen production $M_{H2,d,max}$ (see Table A1). The advantage of LOHC is that it can be transported under ambient conditions (Reuß et al., 2017).

At this point, it must be mentioned that both hydrogenation and liquefication of hydrogen are processes under development. In particular, assumptions about component costs and their scalability to the necessary size for application at wind farms, is uncertain.

### 2.2.2 Tractor and trailers

Different trailers are necessary to transport the hydrogen by truck, depending on the state in which it is stored at the wind farm, as shown in Figure 3. The combination of storage units and trailers that are not of the same type, such as a GH$_2$ tank and a LOHC trailer, is not considered. This is because it would require the necessary infrastructure and auxiliary systems for both technologies and is therefore estimated to be too costly.

For the transport of GH$_2$, tube trailers are used. Due to their high weight, they only have a capacity of approx. 300 $kg_{H2}$. However, current research aims for an improvement of tube trailer capacities up to 1,100 $kg_{H2}$ using alternative materials, which can withstand higher pressures (Adolf et al., 2017). A compressor is needed to increase the pressure of the stored GH$_2$ to the pressure level of the trailers.

LH$_2$ trailers have a much higher capacity of 4,300 $kg_{H2}$, which is due to the higher density compared to GH$_2$ (Reuß et al., 2017). A LH$_2$-pump is required to pump hydrogen from the LH$_2$ storage to the trailer. During transport and unloading approximately 5 % of the hydrogen is lost (Petitpas, 2018).

Conventional petrol trailers are used to transport LOHC-bound hydrogen, resulting in a theoretical capacity of 1,800 $kg_{H2}$ (Reuß et al., 2017). However, during hydrogenation and dehydrogenation, not all hydrogen is processed, again resulting in a loss of approximately 10 % of the trailer's capacity. An additional LOHC-pump is required to fill the trailer. (Petitpas, 2018) The same lifetime is assumed for all trailers. However, the handling time $t_{handling}$ is different for each type of trailer, as shown in Table 2.

A further degree of freedom is the decision on the type of tractor to be used. Today, almost all heavy-duty trucks are diesel-powered (ACEA, 2023). However, both diesel and hydrogen powered tractors are considered, which differ in purchase cost and fuel consumption, as shown in Table 2. The simplified assumption is that the required hydrogen for transport is provided free of charge by the wind farm's hydrogen production. The cost of diesel is estimated at 1.50 € $l^{-1}$. In addition, the labor cost of the truck driver is considered with 35 € $h^{-1}$ (Reuß, 2019). Driver labor costs are calculated based on travel time to the POD

and $t_{handling}$.

A truck access road to the electrolyzer is also required. Road construction costs vary widely depending on local conditions. Based on an expert interview, the cost of an asphalt road, including earthworks, is estimated at 220 € $m^{-2}$ (Kaluk, 2022). The road width is 3 m. Based on the available roads and the position of the electrolyzer $p_{El}$, the road length $l_{road}$ is calculated.

**Table 2: Financial parameters of hydrogen storage, trailers and tractors (Reuß, 2019; Reuß et al., 2017; Petitpas, 2018; Adolf et al.,**
**2016)**

| storage | CAPEX | OPEX | lifetime | / | / |
|---|---|---|---|---|---|
| GH$_2$ | 500 € $kg_{H2}^{-1} \cdot c_{H2,stor}$ | 2 % $a^{-1} \cdot CAPEX$ | 20 a | / | / |
| LH$_2$ | 25 € $kg_{H2}^{-1} \cdot c_{H2,stor}$ | 2 % $a^{-1} \cdot CAPEX$ | 20 a | / | / |
| LOHC | 50 € $kg_{H2}^{-1} \cdot c_{H2,stor}$ | 2 % $a^{-1} \cdot CAPEX$ | 20 a | / | / |
| trailer | CAPEX | OPEX | lifetime | handling time | capacity |
| GH$_2$ | 660,000 € | 2 % $a^{-1} \cdot CAPEX$ | 12 a | 1.5 h | 1,100 kg$_{H2}$ |
| LH$_2$ | 860,000 € | 2 % $a^{-1} \cdot CAPEX$ | 12 a | 3 h | 4,300 kg$_{H2}$ |
| LOHC | 150,000 € | 2 % $a^{-1} \cdot CAPEX$ | 12 a | 1.5 h | 1,620 kg$_{H2}$ |
| tractor | CAPEX | OPEX | lifetime | fuel consumption | |
| diesel | 115,000 € | 12 % $a^{-1} \cdot CAPEX$ | 8 a | 29 l/100 km | |
| hydrogen | 160,000 € | 12 % $a^{-1} \cdot CAPEX$ | 8 a | 6 kg$_{H2}$ /100 km | |

### 2.2.3 Pipeline

An alternative for the hydrogen transport is the use of a hydrogen pipeline. Again, a compressor is required, as shown in Figure 3. In this case, to adjust the pressure level of the electrolyzer to the pressure level of the pipeline. Gas pipelines are divided into transmission, and distribution lines, which operate at different pressure levels. Transmission lines are designed for high
volumes and long distances and operate at high pressure levels, typically above 8.5 MPa, while distribution lines operate at

pressures of 3-4 MPa (Melaina et al., 2013). Comparatively, only small amounts of hydrogen will be produced by decentralized electrolyzers at wind farms, so the parameters of distribution pipelines are used. The outlet pressure of PEMELs varies widely in specifications (Buttler and Spliethoff, 2018). Here, it is assumed that the outlet pressure of the PEMEL is 3 MPa and the pressure in the hydrogen pipeline is 4 MPa.

The cost of the hydrogen pipeline is mainly determined by its radius $r_{H2}$ and length $l_{H2}$ (Mischner et al., 2015). According to Baufumé et al. (2013), no pipeline will be built with a radius smaller than 50 mm. Therefore, the pipeline size is fixed to 50 mm for electrolyzers below 50 MW rated power. It is only variable if $P_{El}$ exceeds 50 MW. $l_{H2}$ is again calculated as the geodesic length between the electrolyzer position $p_{El}$ and the POD position $p_{POD}$ and multiplied by a safety factor $s_{H2}$, which here is 1.4 (Reuß, 2019).

**Table 3 Financial parameters of a hydrogen pipeline (Reuß, 2019)**

| Component | CAPEX | OPEX | lifetime |
|---|---|---|---|
| Pipeline | $292.152 \, € \, m^{-1} \, e^{0.032 \cdot r_{H2} mm^{-1}} \cdot l_{H2}$ | $5 \, € \, m^{-1} a^{-1}$ | 40 a |

## 2.3 Optimization algorithm

In this Section, the implemented optimization algorithm shown in Figure 4 is explained. It is assumed that the entire power consumption of all system components is provided by the wind farm. Therefore, the electricity cost is set equal to the LCoE of the farm. Excess electricity that cannot be used by the electrolyzer because it is running at its rated power $P_{El}$ is fed into the 285 grid (overload operation of the PEMEL is not considered). Therefore, regardless of the electrolyzers capacity utilization $CF_{El}$, the LCoE are assumed to be constant.

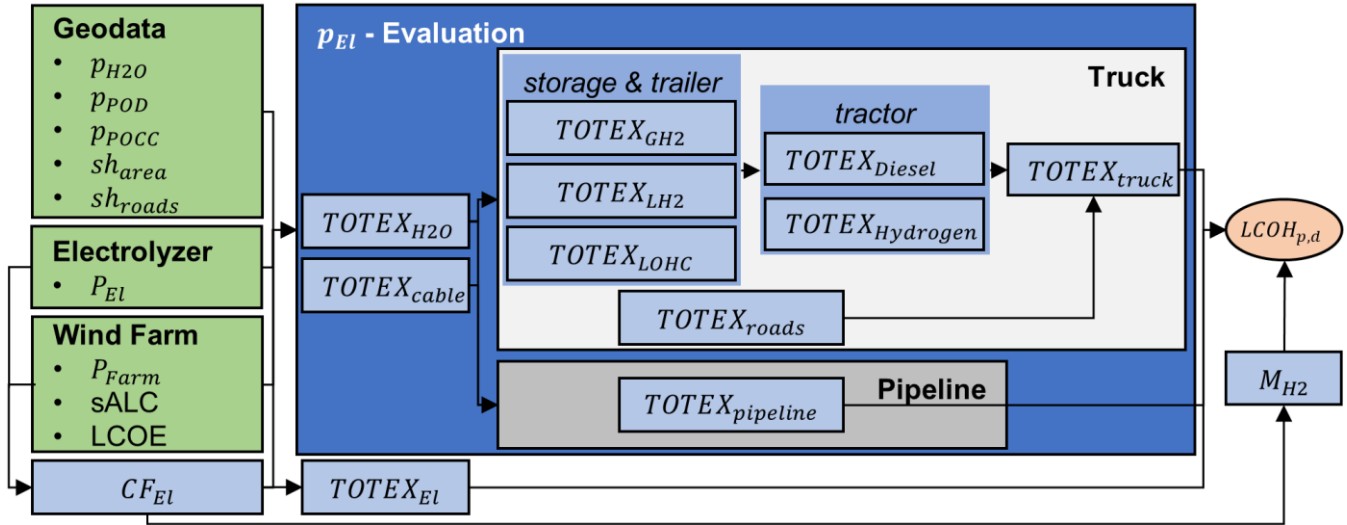

**Figure 4: Illustration of the used optimization algorithm to calculate the electrolyzer position and hydrogen distribution mode to achieve minimum LCoH. Green boxes are input data (fix values), the orange box is output data (target value), all blue boxes are** 290 **automatic calculations, with varying a $p_{El}$ (influencing all values in dark blue box)**

The area to be considered for electrolyzer positioning must be specified as a shapefile $sh_{area}$ in addition to the point data $p_{POD}$, $p_{POCC}$ and $p_{H2O}$ (see Sect. 2.1 and 2.2). $sh_{area}$ is automatically discretized into a point grid resulting in all considered positions $p_{El}$. Existing roads must also be provided as a shapefile $sh_{roads}$. All necessary geodata is processed using the open source software QGIS. Further processing and calculation of all parameters is done in Python. For each possible set of electrolyzer position $p_{El}$ and distribution mode, the resulting TOTEX and finally the LCoH are calculated. The result is the information for which set of $p_{El}$ and hydrogen distribution mode, the LCoH are the lowest.

## 3 Model application and results

In the following, the results of the optimization algorithm introduced in Sect. 2.3 are described. The selected use case is an onshore wind farm in Germany. The positions $p_{POD}$, $p_{H2O}$ and the area considered for the position of the electrolyzer $sh_{area}$ are chosen arbitrary. This also applies to the power curve of the wind farm. It was not provided by the farm operator, but estimated based on wind data.

### 3.1 Use cases

The results of the optimization are shown in Figure 5 (a) and (b). The necessary geodata is created in QGIS and processed in Python. Figure 5 (a) shows the results for a 23.4 MW wind farm, consisting of 13 WTs at 1.8 MW with $v_{cutin}$ of 2.5 $m\ s^{-1}$, $v_{nom}$ of 12.5 $m\ s^{-1}$ and $v_{cutout}$ of 34 $m\ s^{-1}$, combined with a 2 MW electrolyzer, resulting in a $CF_{El}$ of 77 %. Figure 5 (b) shows the results for a wind farm with a rated power $P_{Farm}$ of 58.5 MW. The increase in $P_{Farm}$ could in practice result from a repowering of the wind farm. To investigate the influence of the electrolyzer and wind farm power on the optimal position and distribution mode, the geodata, including the considered area for the electrolyzer $sh_{area}$ and $p_{POCC}$, are kept constant for the use cases. The larger wind farm consist of 13 WTs at a rated power of 4.5 MW with $v_{cutin}$ of 3 $m\ s^{-1}$, $v_{nom}$ of 12 $m\ s^{-1}$ and $v_{cutout}$ of 24.5 $m\ s^{-1}$, combined with a 10 MW electrolyzer, resulting in a $CF_{El}$ of 68 %. The farm-specific sALC and thus $CF_{El}$ for both use cases (a) and (b) is calculated as described in Sect. 2.1.1. The Weibull parameters are specific to the site, with a scale parameter of 7.79 and a shape parameter of 2.13. A wind farm optimization software, introduced by Roscher (2020), was utilized to compute the Weibull parameters.

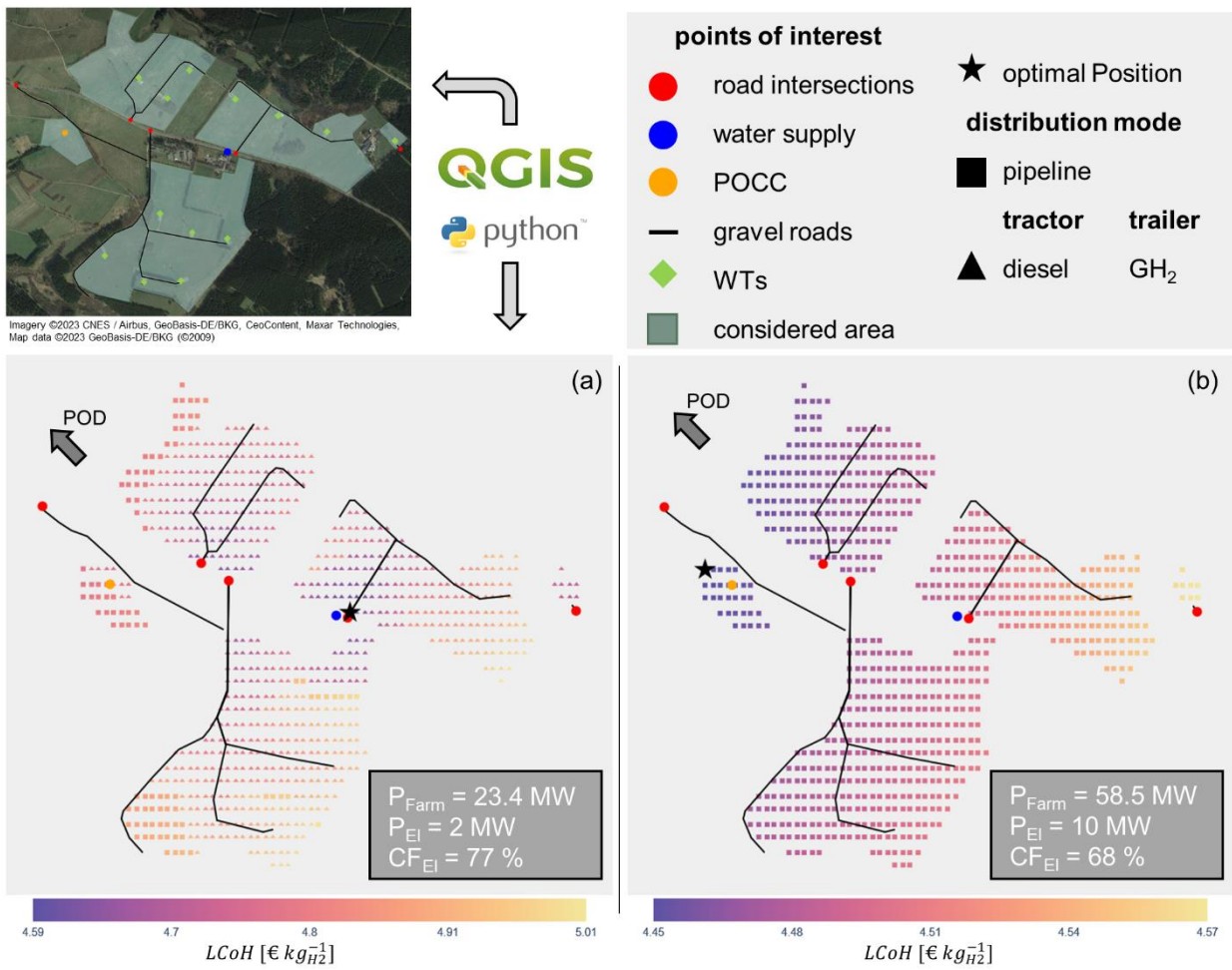

**Figure 5: Optimization results for a German onshore wind farm with LCoE of 5.5 €ct $kW^{-1}$. In both cases (a) and (b), the distance to the POD is approx. 4-6 km, depending on the location $p_{El}$ of the electrolyzer on the wind farm site.**

The result of the optimization tool is a heat map containing all relevant information of the wind hydrogen system. The achievable minimal LCoH ($mLCoH$) is plotted for each electrolyzer position $p_{El}$. This information is indicated by the color of the data point, as shown in the color bar in Figure 5. The $mLCoH_{p,d=d_{min}}$ value of the distribution mode that results in the

lowest cost is plotted, even though there are as many LCoH values for each position $p_{El}$ as distribution modes considered. The marker shape indicates which distribution mode $d$ results in $mLCoH$ at a position $p_{El}$. A black star indicates the combination of position and distribution mode that results in the $mLCoH_{p=p_{El,min},d=d_{min}}$ for the entire area $sh_{area}$.

Based on the calculation results, the achievable mLCoH for the 23.4 MW wind farm combined with a 2 MW electrolyzer is 4.59 € $kg_{H2}^{-1}$. According to Eq. 1, the wind hydrogen-system produces about 283.6 $t_{H2}$ $a^{-1}$. Here, mLCoH is achieved when

a diesel-engine tractor in combination with a GH$_2$-trailer is used. For this use case, the selection of the optimal electrolyzer

position $p_{El}$ on site over the worst position results in a reduction of LCoH of 8.38 %. This applies when comparing the optimal distribution mode for each position.

Figure 5 (a) also shows that LCoH are lower in the vicinity of roads and road intersections with the main road (red dots and black lines) than further away from them, as the road construction costs depend on the required road length $l_{road}$. It is also apparent that mLCoH is achieved for this use case when the electrolyzer is placed in proximity to the position of the water supply $p_{H2O}$ (blue dot). Therefore, a relatively long power cable is required for this wind farm. The reason for this is that the water pipeline costs are higher than the costs for the power cable at low electrolyzer powers $P_{El}$. Depending on the position of $p_{El}$, the POD is only 4 to 6 km away. The impact on the TOTEX and therefore on the LCoH is small as the time required to cover this distance by truck is small. In addition, there are changes in the optimal distribution mode on the considered area for the positioning of the electrolyzer. At the closest locations to the POD (see northwest of the area under consideration in Figure 5), transport by pipeline rather than by truck results in lower LCoH. This is due to the fact that the pipeline length $l_{H2}$ is minimal here, while road construction costs for truck transport are high.

For a specific location $p_{El}$ within the available area $sh_{area}$ Figure 6 showcases and quantifies the above results.

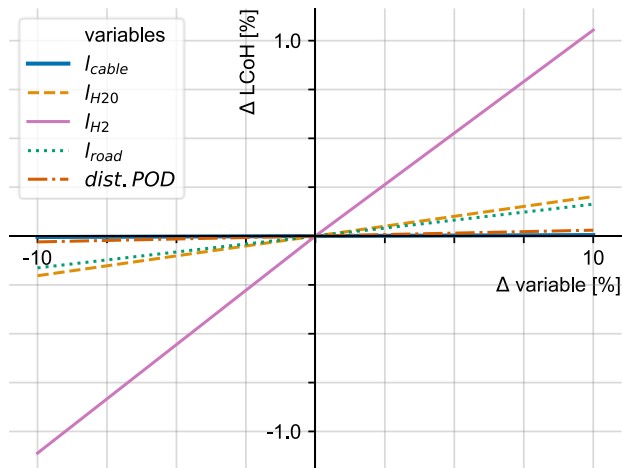

**Figure 6: Sensitivity analysis of impact of varying individual parameters on LCoH for scenario (a). Changes in LCoH due to changes in $l_{H2}$ are relevant only for distribution by pipeline. Changes in distance POD are relevant here for truck transportation.**

For the wind-hydrogen system with $P_{Farm}$ at 58.5 MW combined with a 10 MW electrolyzer, the mLCoH are lower at 4.45 $€\ kg_{H2}^{-1}$. This is due to the 4.4-fold increase in the amount of hydrogen produced per year (1252.18 $t_{H2}\ a^{-1}$) as compared to the small wind-hydrogen system. The 10 MW electrolyzer has a comparatively lower $CF_{El}$. However, the higher amount of produced hydrogen results in a better overall utilization of the required infrastructure, resulting in a LCoH reduction. As a result of the large amount of hydrogen that needs to be transported daily from the wind farm to the POD, pipeline transportation is now the distribution mode resulting in the mLCoH. Figure 5 (b) shows that the optimal electrolyzer position $p_{El}$ is at the northwest edge of the considered area, which leads to the shortest distance pipeline distance $l_{H2}$. The cost of the water pipeline

no longer dominates the optimal position $p_{El}$, as the specific hydrogen pipeline cost per meter is approximately three times higher.

## 3.2 Global LCoH optimum for a wind farm

The results shown in Sect. 3.1 are calculated for a fixed electrolyzer power $P_{El}$. In this case, as shown in Figure 4, $P_{El}$ is an input parameter that is not varied. For the calculation of the global LCoH optimum for a wind farm, $P_{El}$ is now subject to optimization and is therefore also variable.

The optimization algorithm shown in Figure 4 is run for all $P_{El}$ and $P_{Farm}$ ratios $r_{El/Farm}$. $P_{El,max}$ is equal to the rated power of the wind farm $P_{Farm}$, because it is assumed that the electrolyzer is only powered by the wind farm, so $r_{El/Farm}$ is always below 1. This results in the minimum LCoH that can be achieved for a combination of wind farm and POD, referred to as $mLCoH_{p,d,P_{El}}$.

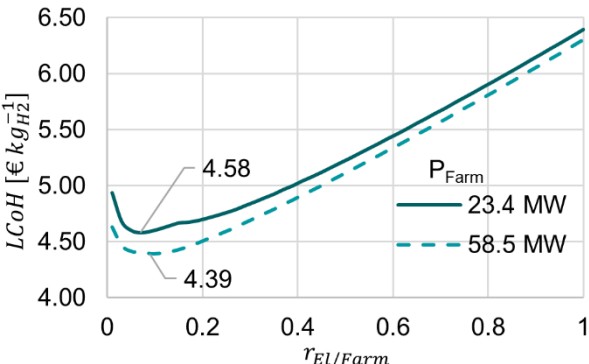

Figure 7: Results of the global LCoH optimization for two use cases

As shown in Figure 7, $mLCoH_{p,d,P_{El}}$ is obtained at an $r_{El/Farm}$ of about 0.1. A low $r_{El/Farm}$-value results in a high electrolyzer capacity utilization $CF_{El}$, see also Figure 2. Thus, for the use case shown in Figure 5 (a) and discussed in Sect. 3.1, the electrolyzer is therefore almost optimally sized with an $r_{El/Farm}$-value of 0.085, while the rate of 0.17 is above the optimum for use case (b), resulting in LCoH higher than mLCoH. For larger values of $r_{El/Farm}$ respectively smaller $CF_{El}$-values, the LCoH increase almost linearly. This is mainly due the infrastructure supplying the electrolyzer being designed for its rated power $P_{El}$. Consequently, the infrastructure costs scale linearly with $P_{El}$. The cost of the electrolyzer also increases with its size. For almost all components, the OPEX are also based on their CAPEX. The design and thus the cost of the infrastructure for on-site hydrogen storage and distribution is also based on $P_{El}$ (see Sect. 2.2.). The TOTEX of a hydrogen system with a larger electrolyzer, but lower $CF_{El}$ increase more than the mass of hydrogen it can produce annually $M_{H2,a}$. According to Eq. 1, this results in an increase in LCoH.

For smaller $r_{El/Farm}$, the LCoH also increase, in this case almost exponentially. In case $r_{El/Farm}$ falls below a certain value, the utilization of the electrolyzer $CF_{El}$ does not increase any further, since wind farms typically do not produce electricity for

a certain time period of the year (cf. Sect. 2.1.1). As $P_{El}$ decreases, fixed CAPEX such as road construction costs do not decrease. TOTEX therefore decrease at a lower rate than $M_{H2,a}$, resulting in higher LCoH. In any case, if the value $r_{El/Farm}$ is too high or too low, some infrastructure components will not be used optimally.

### 3.3 Distribution mode analysis

For further analysis, the dependence of the distribution modes on the distance to the POD and the daily hydrogen production are investigated. This allows sensitivity analysis of input parameters for selecting different distribution modes. All parameters except the distance to the POD and $P_{El}$ are constant, including the electrolyzer position $p_{El}$ on the wind farm site. In particular, $CF_{El}$ is kept constant at 70 % which implies that $P_{Farm}$ and/or the wind farms FLH must increase with $P_{El}$. Figure 8 shows the results of the distribution mode analysis. The plot shows only the LCoH for the distribution mode, which results in the mLCoH for a daily hydrogen production $M_{H2,d}$, distance to POD combination. The black line indicates where the LCoH for pipeline transport are higher than the LCoH for truck transport, or vice versa. For high hydrogen mass flows, but short distances, the pipeline is the most economical distribution mode. Up to an electrolyzer power of 50 MW, the pipeline cost is only dependent on its length $l_{H2}$ (see Sect. 2.2.3). Due to the low capacity of hydrogen trailers (see Table 2) multiple trailers are required for high daily hydrogen production $M_{H2,d}$, increasing the LCoH.

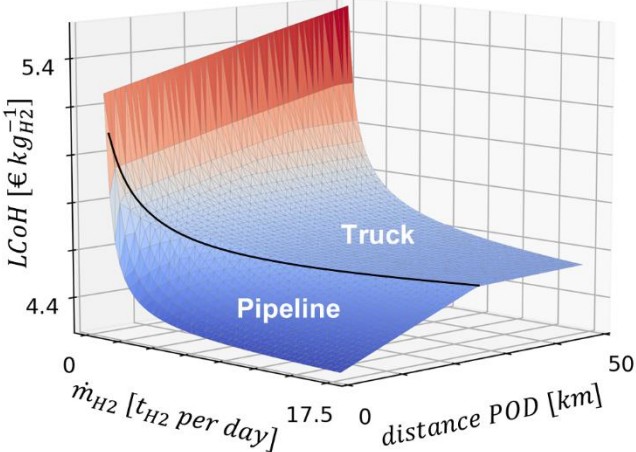

**Figure 8: Distribution mode analysis results for a minimum $P_{El}$ of 1 MW and a maximum $P_{El}$ of 50 MW. The $CF_{El}$ is constant at 70 % for each datapoint**

However, for low daily hydrogen productions and short distances, as well as a high daily production and long distances to the POD, transportation by truck is cheaper than by pipeline, as shown in Figure 8. For long distances, the high cost of pipelines exceeds the cost of truck transportation.

Figure 9 provides a more detailed analysis of the most favorable distribution mode depending on distance to the POD and daily hydrogen production. Therefore, results are shown for two different exemplary diesel fuel prices and excluding a pipeline as

a distribution mode. Instead of the LCoH, the color indicates the different distribution modes, including the different tractor and trailer combinations. For wind farm sites where construction of a hydrogen pipeline is not possible or not permitted, consideration of the operating windows without the pipeline is relevant.

As shown in Figure 9 (a) and discussed before, large hydrogen mass flows and short distances to the POD result in the pipeline
being the most economical distribution mode. Transportation by a $GH_2$-trailer is best suited for long distances and small amounts of hydrogen.

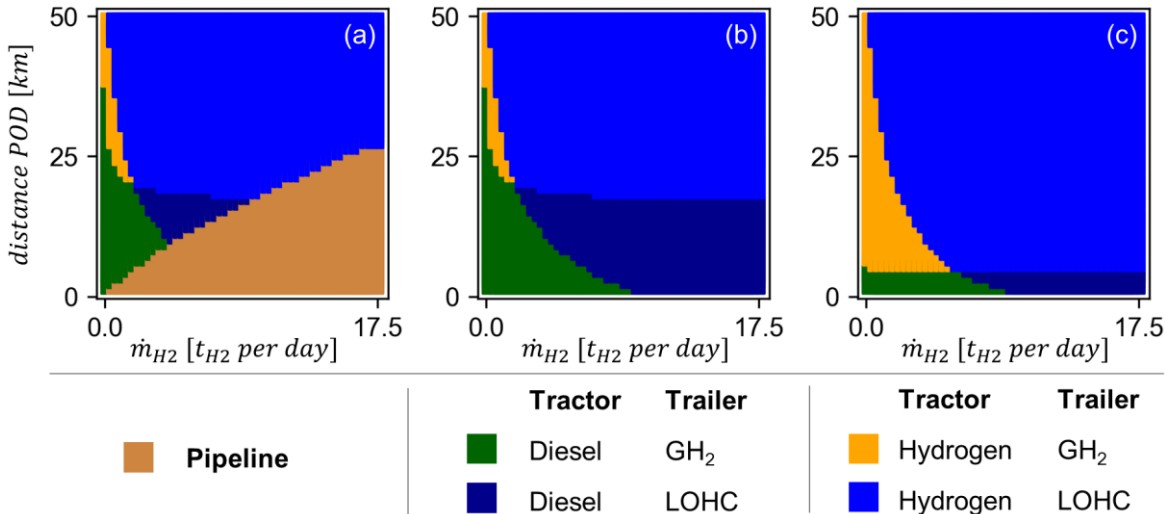

**Figure 9: Detailed distribution mode analysis for (a) a diesel price of 1.5 $\text{€ } l^{-1}$, considering a pipeline, (b) a diesel price of 1.5 $\text{€ } l^{-1}$, not considering a pipeline and (c) a diesel price of 2.5 $\text{€ } l^{-1}$, not considering a pipeline**

Transportation in a LOHC trailer is only feasible for wind-hydrogen systems with a larger daily hydrogen production. Although the LOHC trailer is the least expensive of the trailers considered (see Table 2), it requires expensive additional infrastructure (see Figure 3). However, the higher capacity and lower price of the trailers makes the investment in the additional infrastructure economically viable at a certain amount of hydrogen produced per day.

For long distances to the POD, the additional investment in more expensive hydrogen tractors is reasonable. This is because
there is no additional cost included in the model for hydrogen consumed by trucks. It is assumed that the hydrogen consumption of the trucks is covered by the production of the wind farm, so that the hydrogen price is equal to the LCoH. This is a simplification, as additional infrastructure is required for hydrogen refueling at the wind farm and the hydrogen used cannot be sold, which must be taken into account in any economic analysis. Still, the increase in diesel price from Figure 9 (b) to (c) shows that the operating window for diesel tractors can be reduced by increasing fuel prices.

No combination of considered distance to POD and hydrogen mass flow results in mLCoH for the use of $LH_2$ trailers. While $LH_2$ trailers have the largest hydrogen capacity, they are also the most expensive trailers (see Table 2). In addition, infrastructure components are required on site to load and unload $LH_2$ trailers.

## 4 Discussion and future work

In this paper, the influence of the optimal electrolyzer position $p_{El}$ at a wind farm site in combination with the optimal hydrogen distribution mode on the LCoH has been discussed. Therefore, a novel optimization method based on analytical equations has been developed. The implemented methodology leads to transparent and reproducible results for LCoH, which are in line with the LCoH for green hydrogen as reported in the literature (Ajanovic et al., 2022). Hofrichter et al. (2023b), who conducted a study on the optimal ratio between electrolyzer and wind farm size, calculated mLCoH of 2.53 € $kg_{H2}^{-1}$. The lower LCoH is partly due to a lower WACC and LCoE considered, and partly due to the fact that hydrogen transportation costs are not considered. Hofrichter et al. (2023b) conclude that a higher optimal $r_{El/Farm}$ results in lower LCoH, and that higher installed capacities of renewables lead to lower LCoH, which is in line with the results of this work. Since infrastructure components are sized based on electrolyzer capacity, LCoH increase for increasing $r_{El/Farm}$ in this study. In their review of 18 papers, Bhandari and Shah (2021) concluded that the LCoH for decentralized hydrogen production with PEMEL is 1.90-7.56 € $kg_{H2}^{-1}$. The LCoH calculated in this paper fall within this range. As shown here, the electrolyzer position $p_{El}$ and the distribution mode have a significant impact on the LCoH of a wind-hydrogen system combined with a POD. Depending on the distribution mode, the main parameters affecting $p_{El}$ are the pipeline length $l_{H2}$ and the expenses for roads and water pipes. It is now possible to calculate the optimal $r_{El/Farm}$ to achieve mLCoH for an individual wind farm site, considering local wind conditions and WT types. For the investigated use case of a 23.4 MW wind farm the optimal ratio lies at around 10 %. The method requires minimal data input. Therefore, it is easily transferable to other onshore wind farm sites.

The impact of technological advances such as an increased electrolysis efficiency, component lifetime extension or cost reduction on the LCoH of a wind-hydrogen system can be analyzed by varying the input parameters. This has been done for the example of a rising diesel price. This allows future-proof decisions to be made as early as in the planning phase of a project. The developed method thus generates added value for both, research and industry. Wind farm operators and planners can use the method to obtain a sound estimate of the achievable LCoH for a wind-hydrogen system. The results can be used for detailed planning. In science, the method can be further applied to hybrid power plants, consisting of photovoltaic, batteries and wind farms, to further reduce green hydrogen cost. In addition, the impact of technological advances on LCoH can be assessed.

Wind-hydrogen systems are complex and consist of many components. Several components are still in the early stages of development. Estimates of costs, efficiencies, and other system parameters are therefore subject to uncertainty. In addition, the design of the overall wind-hydrogen system requires the use of physically simplifying assumptions. Various model limitations have already been described in Sect. 2. In particular, the efficiency of the electrolyzer is assumed to be constant regardless of load. Hofrichter et al. (2023a) show that the efficiency of a PEMEL is higher at partial load compared to full load. This could potentially increase the optimal $r_{El/Farm}$. However, due to infrastructure components being sized based on the electrolyzer capacity, the effect will be limited. The AEP of the wind farms is calculated using site specific Weibull parameters and the turbines' power curves. The AEP is slightly overestimated due to neglect of wake effects, resulting in small uncertainties in an optimal $r_{El/Farm}$. A detailed investigation of each component is beyond the scope of this work and would increase the required

computational effort. Currently, on an AMD Ryzen 7 Pro 6850U with 2.7 GHz with 32 GB of RAM, the optimization is performed in less than 900 s for typical wind farm sites. Although the results of the method will change as the input parameters are modified, the dependencies shown for LCoH of wind-hydrogen systems on electrolyzer position and distribution modes remain. However, the detailed design of a wind-hydrogen-system must be site-specific.


The implemented methodology offers several opportunities for future research, some of which are highlighted below.

According to Eq. 1 and 5, the LCoH are inversely proportional to $CF_{El}$ (see Eq. 8). Currently, the electricity for the electrolysis process is only provided by the wind farm. There is potential to increase $CF_{El}$ by using power from both, wind and photovoltaic systems, as there is little competition for time-resolved power feed-in from both technologies (Gerlach et al., 2011).

$$LCoH_{p,d} = \frac{\text{TOTEX}_{p,d}}{M_{H2,a}} \sim \frac{1}{CF_{El}} \qquad (8)$$

To maximize the utilization of the electrolyzer $CF_{El}$, additional electricity can be purchased from the grid. However, the price of electricity then depends on the market price of electricity at the time of consumption. Therefore, the additional purchase of electricity during periods of low electricity prices may further reduce the achievable LCoH. In addition, FLH and thus the $CF_{El}$ can be increased by integrating PV systems and battery storage into the system. The impact on LCoH needs to be investigated,

considering detailed infrastructure costs, as in this study.

Both, the power grid and road layout for a wind farm are usually planned in an early design phase, when the optimal WT positions are evaluated (Roscher, 2020). Since additional power cables and roads need to be built for the electrolyzer, it may have an impact on the overall wind farm grid and road layout. Instead of optimizing the hydrogen system sequentially, the optimization process needs to be integrated into a wind farm optimizer.

As described in Sect. 2, the optimization method introduced in this paper aims to achieve mLCoH for a wind-hydrogen-system. Blickwedel et al. (2021) introduce the metric Levelized Revenue of Electricity (LRoE). Unlike LCoE or LCoH, it does not measure the costs of a plant, but its revenue. For a wind-hydrogen plant, designing the system to maximize the LRoE is the next step. A controller must be developed, considering the electricity and hydrogen market prices. This controller must decide when to produce hydrogen or feed electricity into the grid.

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

## A Appendix

**Table A1: Financial parameters of the support infrastructure, $f$ is 2.5 for a pipeline compressor and 3 for a compressor used for a trailer (Reuß, 2019). Depending on the pressure level of the pipeline or GH₂ trailer, $P_c$ is calculated.**

| component | CAPEX | OPEX | lifetime | efficiency |
|---|---|---|---|---|
| compressor | $15{,}000\,€\,kW^{-1}\cdot P_c^{0.6098}\cdot f$ | $4\,\%\cdot a^{-1}$ | 15 a | 99.5 % |
| liquefier | $105\text{ Mio. }€\cdot\left(\frac{M_{H2,d,max}}{50\cdot t_{H2}}\right)^{0.66}$ | $4\,\%\cdot a^{-1}$ | 20 a | 96.45 % |
| evaporator | $3{,}000\,€\cdot\frac{M_{H2,d,max}}{t_{H2}}$ | $3\,\%\cdot a^{-1}$ | 10 a | 100 % |
| hydrogenation | $40\text{ Mio. }€\left(\frac{M_{H2,d,max}}{300\cdot t_{H2}}\right)^{0.66}$ | $3\,\%\cdot a^{-1}$ | 20 a | 99 % |
| dehydrogenation | $30\text{ Mio. }€\left(\frac{M_{H2,d,max}}{300\cdot t_{H2}}\right)^{0.66}$ | $3\,\%\cdot a^{-1}$ | 20 a | 99 % |
| LH2-pump | $30{,}000\,€\cdot\frac{M_{H2,d,max}}{t_{H2}}$ | $3\,\%\cdot a^{-1}$ | 10 a | 100 % |
| LOHC-pump | $500\,€\cdot\frac{M_{H2,d,max}}{t_{H2}}$ | $3\,\%\cdot a^{-1}$ | 10 a | 100 % |

## Code/Data availability

No data other than that given in the paper is required to produce the presented results. The source code is not published.

## Author contribution

The concept and method of the present study were developed by Tom Rathmes and Thorsten Reichartz. Tom Rathmes performed the initial implementation of the software, which was extended and improved by Thorsten Reichartz and Lucas

Blickwedel. Thorsten Reichartz performed the initial text creation. Lucas Blickwedel, Ralf Schelenz and Georg Jacobs were responsible for supervision, revision, and final approval.

All authors have read and agreed to the published version of the paper.

## Competing interests

The authors declare that they have no conflict of interest.