# Peer review of "Optimal Position and Distribution Mode for On-Site Hydrogen Electrolyzers in Onshore Wind Farms for minimal LCoH"

_Wind Energy Science, 2023_

## Author Response (AR1)

**Information how to read the document**

- RC: Referee Comment
  - Pages and lines in the author comments refer to the manuscript submitted before the review process, as we did not want to change author comments here.
- AR: Authors Response
  - To avoid confusion, we have removed the line references in our responses that we mentioned in our author comments, as these referred to the unmarked revised manuscript, and we now include them in the author changes.
- AC: Authors Changes
  - Parts indicated in blue here were added, parts indicated red and which are crossed out were deleted – all lines and page numbers refer to the lines in the marked-up manuscript for easy trackability.

**RC1**

*Specific comments & crucial assumptions:*

| RC | The manuscript encompasses relevant parameters related to infrastructure planning – and is indeed relevant for system planners realizing these wind-hydrogen systems. However, there are some crucial assumptions, which might impact the results, and the manuscript could benefit from discussing these assumptions and potential impacts (e.g. based on findings in the existing literature). Among the crucial assumptions are, e.g., 1) efficiency changes depending on operation, … |
|---|---|
| AR | Regarding the issue of the crucial assumption on efficiency changes in operation, we have now added a brief discussion on the implications of this in Sect. 4 of the manuscript. Future development of our method will address and incorporate efficiency changes due to partial load. |
| AC | Page 19, lines 467-469 |
|  | In particular, the efficiency of the electrolyzer is assumed to be constant regardless of load. Hofrichter et al. (2023a) show that the efficiency of a PEMEL is higher at partial load compared to full load. This could potentially increase the optimal $r_{El/Farm}$. However, due to infrastructure components being sized based on the electrolyzer capacity, the effect will be limited. |

| RC | … 2) wake effect losses are neglected, … |
|---|---|
| AR | Regarding the comment on wake effect losses: We clarified in a comment above how the sorted annual load curve was determined for the wind farms and in the manuscript. The authors agree that this is a strong simplification. However, in this particular work, it is done in order to minimize the modeling effort, since the focus is not on the AEP calculation, but on the modeling of the hydrogen system, including hydrogen distribution. It has been stated in Sect. 2.1.1 that historical SCADA data should be utilized if available. If not, wake models such as the wake model introduced by Katic (https://orbit.dtu.dk/en/publications/a-simple-model-for-cluster-efficiency) can be used. It is important to note that, as per the Katic model, the effect of wake becomes quadratically stronger at higher wind speeds. Consequently, the margin of error for the results obtained in this study becomes more significant for larger electrolyzer/farm ratios and higher-rated electrolyzer powers. As shown in the left panel of Figure 2, small electrolyzer capacities operate far below the power output of the wind farm for the majority of the year. When they do operate at similar levels, it is only during periods of low wind speeds, resulting in relatively low wake impacts, according to Katic. We show that low electrolyzer/farm ratios are optimal, where the results are least affected by the error caused by ignoring the wake. However, we agree with the reviewer's critique and have added a brief discussion of the impact in Sect. 4 of the manuscript. |
| AC | Page 6, lines 154-159 & Page 19, lines 469-472 |
|  | To obtain the sALC of a wind farm, the curve is multiplied by the number of turbines in the farm.  This simplification is assumed to be sufficient  for the subject of this work. However, a more accurate sALC, considering wind turbine (WT) positions and wake effects can be generated, for example, using the methodology described by Shapiro et al. (2019) or one of the wake models discussed |

|  |  |
| --- | --- |
|  | by Brusca et al. (2018). As shown in Figure 2, based on the sALC and the rated power of the electrolyzer $P_{El}$, the equivalent $FLH_{El}$, $W_{El}$ and thus $CF_{El}$ is computed.

The AEP of the wind farms is calculated using site specific Weibull parameters and the turbines' power curves. The AEP is slightly overestimated due to neglect of wake effects, resulting in small uncertainties in an optimal $r_{El/Farm}$.  A detailed investigation of each component is beyond the scope of this work and would increase the required computational effort. |

| RC | … 3) potential excess heat revenues, … |
| --- | --- |
| AR | Regarding potential excess heat revenues: This study was solely focused on cost approximation and minimization of hydrogen production. Utilizing waste heat from the electrolysis process can potentially improve the overall system efficiency and have a positive impact on the economic efficiency of the system. However, it does not affect the LCoH, which is minimized in this study. Therefore, it is not considered. Nonetheless, in a detailed profitability analysis of a hybrid wind farm, income from waste heat utilization should be considered. This would also require modeling all necessary infrastructure components for heat transport and exchange |
| AC | - |
|  | - |

| RC | … 4) availability of using grid electricity and/or hydrogen production from wind-solar PV systems. |
| --- | --- |
| AR | The question on the availability of grid electricity has been addressed within Sect. 4 with regards to reducing LCoH (see Eq. 8). However, we included an extra statement to indicate that the availability of additional electricity from photovoltaic for hydrogen production may lead to a further reduction in LCoH, due to the complementary power feed-in of photovoltaic and wind energy. |
| AC | Page 20, lines 480-481 |
|  | Currently, the electricity for the electrolysis process is only provided by the wind farm. There is potential to increase $CF_{El}$ by using power from both, wind and photovoltaic systems, as there is little competition for time-resolved power feed-in from both technologies (Gerlach et al., 2011). |

| RC | Also please, reflect upon the obtained results (LCoH) in related to other existing studies, and their potential uncertainties. |
| --- | --- |
| AR | Also as per your suggestion, a section has been added in Sect. 4 where our findings on LCoH are reflected in relation to previous studies. |
| AC | Page 19, lines 443-450 |
|  | The implemented methodology leads to transparent and reproducible results for LCoH, which are in line with the LCoH for green hydrogen as reported in the literature (Ajanovic et al., 2022). Hofrichter et al. (2023b), who conducted a study on the optimal ratio between electrolyzer and wind farm size, calculated mLCoH of 2.53 € $kg_{H2}^{-1}$. The lower LCoH is partly due to a lower WACC and LCoE considered, and partly due to the fact that hydrogen transportation costs are not considered. Hofrichter et al. (2023b) conclude that a higher optimal $r_{El/Farm}$ results in lower LCoH, and that higher installed capacities of renewables lead to lower LCoH, which is in line with the results of this work. Since infrastructure components are sized based on electrolyzer capacity, LCoH increase for increasing $r_{El/Farm}$ in this study. In their review of 18 papers, Bhandari and Shah (2021) concluded that the LCoH for decentralized hydrogen production with PEMEL is 1.90-7.56 € $kg_{H2}^{-1}$. The LCoH calculated in this paper fall within this range. As shown here, … |

| RC | An overview of existing literature must be integrated to highlight the novelty of the paper and/or to align with other studies. The current statement p. 2, line 40-41 "No relevant literature could be identified that quantifies leverage or describes how to generate optimal solutions for the given case", could therefore be reconsidered. |
| --- | --- |
| AR | As recommended, we included a review of the existing literature and removed the statement about the lack of relevant literature. Our study's contributions are also discussed now in relation to the existing studies. |
| AC | Page 2-3, lines 43-76 |

Numerous studies have addressed the subject, including Hofrichter et al. (2023b), who investigated the optimal power ratio of electrolyzers and renewable energy sources. Their analysis covered wind farm sites characterized by varying full load hours (FLH), but did not consider hydrogen transportation costs nor on-site electrolyzer positioning. Similarly, Schnuelle et al. (2020) and Benalcazar and Komorowska (2022) take the macroscopic approach of evaluating sites based on FLH, neglecting hydrogen transport and microscopic assessments that include ancillary infrastructure requirements such as existing roads and water pipelines. In their study on hydrogen production from floating offshore wind, Ibrahim et al. (2022) adress the transportation of energy to shore in the form of hydrogen or electricity, considering the central role of energy distribution within energy systems. The study focuses on offshore wind to hydrogen, which limits its transferability to onshore farms. Sens et al. (2022) investigate the ideal locations on a continental and regional scale for hydrogen production from wind and solar to provide hydrogen to Germany, including hydrogen transportation costs, but they only consider pipeline transportation as they focus on large quantities of produced hydrogen. The authors also made it explicit that they excluded transmission costs for electricity and water on-site. While other studies have also analyzed the costs of the necessary infrastructure for hydrogen production and transportation at the macroscopic level (Yang and Ogden, 2007; Reuß, 2019; Correa et al., 2022), transferable models for a specific cost analysis at the wind farm level, including detailed site-specific infrastructure, electrolyzer positioning and transport mode optimization, are not available. This  publication aims to address and fill that gap by answering the foll**ow**ing research question and sub questions:

- **To what extent can wind farm operaters and developers reduce the LCoH of green hydrogen produced at wind farm sites?**

    o    What are relevant influencing factors on LCoH of on-site wind hydrogen systems?

    o    How  can those be modelled ?

    o    What level of LCoH can be achieved, and what is the ideal electrolyser/wind farm power ratio to achieve this minimum, taking into account hydrogen transport and all required infrastructure at a specific wind farm site?

 Despite the environmental benefits of green hydrogen, its production costs must be reduced in order to compete with grey hydrogen (Ajanovic et al., 2022). Decentralized hydrogen production brings its own challenges, such as the need to position electrolyzers on wind farm sites, establish deionized water and electricity supply, and transport the hydrogen off-site.

| RC | Many of the input parameters are well-documented in the manuscript, however, how did you obtain the wind profiles for the two different onshore wind farms (23.8 MW, and 58.5 MW)? |
|---|---|
| AR | The wind profiles for the wind farm were obtained using our in-house wind farm optimizer (10.18154/RWTH-2020-03444). It uses open access data from the German Weather Service (DWD https://opendata.dwd.de/climate_environment/CDC/grids_germany/multi_annual/ wind_parameters/resol_1000x1000/). Our calculations resulted in site-specific Weibull parameters (wind profile) with a scale parameter A of 7.79 and a shape parameter k of 2.13 at 80 m above ground. The existing wind farm has 13 turbines installed, each rated at 1.8 MW, with a cut-in wind speed of 2.5 m/s, a nominal wind speed of 12.5 m/s, and a cut-out wind speed of 34 m/s. We used a corresponding power curve. We have now clarified this in the manuscript and added the necessary information to calculate the sorted annual load curve (sALC). We have also changed each 23.8 MW in the manuscript to 23.4 MW (13*1.8 MW) (including Figure 5 and Figure 7). This was a typographical error, and all data based on it have been calculated using the correct value of 23.4.
 The sALC for the larger rated wind farm was obtained by using the power curve of 13 wind turbines, each rated at 4.5 MW, using the same site-specific Weibull parameters, but a cut-in wind speed of 3 m/s, a nominal wind speed of 12 m/s, and a cut-out speed of 24.5 m/s. The relevant parameters for calculation of LCoH are the sALC, possible positions of the electrolyzer (available area) and the POCC, as shown in |

|    |    |
|----|----|
|    | Figure 4. The position of turbines is insignificant except for calculating the sALC. To ensure comparability of the use cases for the two wind farms with a rated power of 23.4 MW and 58.5 MW, the available area and the POCC were kept constant. This has now also been clarified in the manuscript. |
| AC | Page 12-13, lines 317-330, Page 13, Figure 5, Page 16, Figure 7 |
|    | The results of the optimization are shown in Figure 5 (a) and (b). The necessary geodata is created in QGIS and processed in Python. Figure 5 (a) shows the results for a 23. 4 MW wind farm, consisting of 13 WTs at 1.8 MW with $v_{cutin}$ of 2.5 $m\ s^{-1}$, $v_{nom}$ of 12.5 $m\ s^{-1}$ and $v_{cutout}$ of 34 $m\ s^{-1}$, combined with a 2 MW electrolyzer, resulting in a $CF_{El}$ of 77 %. Figure 5 (b) shows the results for a wind farm with a rated power $P_{Farm}$ of 58.5 MW. The increase in $P_{Farm}$ could in practice result from a repowering of the wind farm. To investigate the influence of the electrolyzer and wind farm power on the optimal position and distribution mode, the geodata, including the considered area for the electrolyzer $sh_{area}$ and $p_{POCC}$, are kept constant for the use cases. The larger wind farm consist of  13 WTs at a rated power of 4.5 MW with $v_{cutin}$ of 3 $m\ s^{-1}$, $v_{nom}$ of 12 $m\ s^{-1}$ and $v_{cutout}$ of 24.5 $m\ s^{-1}$, combined with a 10 MW electrolyzer, resulting in a $CF_{El}$ of 68 %. The farm-specific sALC and thus $CF_{El}$ for both use cases (a) and (b) is calculated as described in Sect. 2.1.1. The Weibull parameters are specific to the site, with a scale parameter of 7.79 and a shape parameter of 2.13. A wind farm optimization software, introduced by Roscher (2020), was utilized to compute the Weibull parameters.  |

|    |    |
|----|----|
| RC | **Furthermore, it is a bit unclear how the electrolyzer capacity at 2 MW and 10 MW, respectively to the two onshore wind farms, were determined?** |
| AR | Regarding the question on how we determined the electrolyzer capacity at 2 MW and 10 MW for the two use cases described in Sect. 3.1, these are two exemplary combinations of a wind farm and an electrolyzer. Figure 7 provides a discussion on every possible combination between electrolyzer and wind farm we considered. However, it is not practical to display every calculated combination. The results in Figure 5 present interesting cases regarding the switching favorable distribution modes within the possible electrolyzer positions. In addition, by comparing the results for the use cases shown, it is clear that the position of the electrolyzer depends on the capacity of the electrolyzer and the amount of hydrogen produced. |
| AC | - |
|    | - |

|    |    |
|----|----|
| RC | **In section 4 "Discussion and further work" please, search within the literature if hybrid power plants (p 17, line 414-415) and/or purchasing electricity from the grid (p. 18, line 431-428) have already been modelled and studied. If so, please reformulate the parts related to future work.** |
| AR | Regarding the comment on line 414-414: While other studies have examined the impact of grid and hybrid power, we believe this is a critical next step in our LCoH calculation tool. This is because it considers the entire site-specific infrastructure and transport mode. We now point out this limitation in the manuscript, but generally keep the reference to future work in the paper for the reasons mentioned above. |
| AC | Page 20, lines 486-487 |
|    | The impact on LCoH needs to be investigated, considering detailed infrastructure costs, as in this study.  |

*Technical Corrections:*

|    |    |
|----|----|
| RC | **g. p3. Line 93: Be consistent with LCoE (is the E for electricity or energy)?** |
| AR | The abbreviation "E" stands for electricity in LCoE and has been corrected in line 35. It is not reintroduced a second time as well now. |
| AC | Page 2, line 36 & Page 4, line 115 |
|    | One possibility to reduce LCoH is to further reduce the Levelized Cost of  Electricity (LCoE) of wind turbines… |

| | The LCoE and the generation |
|---|---|

| RC | 3, line 95, use hydrogen throughout the text (not H2). |
|---|---|
| AR | Line 95: H2 has now been changed to hydrogen. |
| AC | Page 4, line 117 |
| | This does not apply for components necessary for unloading the hydrogen trailers and converting hydrogen back into a gaseous state . |

| RC | 8. Three times a Table is references, but without the Table "number". |
|---|---|
| AR | The proper Table references are now given. |
| AC | Page 9, lines 234, 239, 246 |
| | - |

AR: Also, all additional typographical errors have been corrected.

| RC | 9, line 235 and 240, reference should be within the sentence. |
|---|---|
| AC | Page 10, line 262 |
| | approximately 5 % of the hydrogen is lost (Petitpas, 2018). |
| RC | 9, line 237, references are repeated three times. |
| AC | Page 10, line 264 |
| | resulting in a theoretical capacity of 1,800 $kg_{H2}$ (Reuß et al., 2017).  |
| RC | 9, line 295. It seems that "-" should be "." |
| AC | Page 12, line 320 |
| | - |

**RC2**

*Major Questions:*

| | |
|---|---|
| **RC** | **Line 123: Is the electrolyzer using all of the energy from the wind farm that it can? Is it assumed that selling hydrogen is more profitable than electricity? Please state your assumptions here.** |
| **AR** | The electrolyzer consumes all the electricity it can, i.e. a 1 MW electrolyzer will consume 1 MWh of electricity in 1 hour if the amount of electricity produced by the wind farm in that hour is equal to or greater than 1 MWh, and if the power output is always greater than 1 MW in that hour. This is also shown in Eq. (4), where the available energy $W_{El}$ is used to calculate $CF_{El}$. (please also see the comment on your comment on Line 292)

No assumptions are made about the profitability of selling hydrogen or electricity. The objective is to minimize the Levelized Cost of Hydrogen (LCoH), as given in Eq. (1), which is a pure cost-based question. The question of when to sell electricity to the electricity grid and when to use it for hydrogen production to optimize the revenue and profit of a wind-hydrogen system is a complex control and energy management issue for hybrid farms. This has been addressed within other work. As stated in our future work section, it should be considered to combine our method with those methods, to calculate an optimal Levelized Revenue of Hydrogen in the future. However, it is not certain which parameter will ultimately be the more important one to optimize, due to uncertainty about market trends regarding electricity and hydrogen prices. |
| **AC** | Page 5, line 149 |
| | $W_{El}$ depends on the amount of electricity generated by the connected wind farm. This energy is defined as the Annual Energy Production (AEP). It is assumed that the difference between the AEP and $W_{El}$ is fed to the electricity grid. |

| | |
|---|---|
| **RC** | **Lines 283-284: How is the optimization calculated? Is it trying out all of the combinations? How would this scale with land area, for example?** |
| **AR** | Currently, the optimization algorithm is implemented in such a way that it tries all combinations, so that the computation time T(n) scales with O(n), where n is the land area, since the evaluated points double with double land area. So far, the authors have not encountered any problems with the computational time, since only linear equations have to be solved for the discrete point evaluation, which does not require much computing power. In case of computational problems due to large areas to be analyzed, the grid resolution of the discretized points can be adjusted to reduce the number of points to be analyzed. The computation time T(m) would be reduced by O(1/m²), where m is the distance between the grid points. The resolution could be increased again for areas with promising LCoH determined in this way. |
| **AC** | - |
| | - |

| | |
|---|---|
| **RC** | **Line 292: Might be good to reference later in the paper where you do a study for optimum electrolyzer size. Also, is the extra electricity produced sold to the grid? Is this included in the calculation? How does this affect the profitability of the wind farm?** |
| **AR** | We agree with this suggestion and have added a reference to the two use cases discussed in Sect. 3.1 to Sect. 3.2 and put their electrolyzer/farm ratios in the context of the optimal electrolyzer size study.

It is assumed that any extra electricity not used by the electrolyzer is sold to the grid, but not meaning the profitability is considered, as already clarified in the response to the comment to line 123: The economics of a wind-hydrogen farm are not evaluated here. But in a way, that not using the excess electricity not used by the electrolyzer (equal to $AEP - W_{El}$, according to Eq. (4)) would negatively affect the Levelized Cost of Electricity (LCoE) of the wind farm, which are kept constant during the optimization (see Figure 4). This is due to the way LCoE are usually calculated (discounted TOTEX of a wind farm divided by its AEP). Not using the remaining electricity would therefore be equivalent to reducing the AEP of the wind farm, which would result in an increasing LCoE and therefore LCoH.

To clarify on that, we added an additional sentence to Eq. (4), stating that it is assumed that the difference between the AEP and $W_{El}$ is fed to the power grid. |
| **AC** | Page 5, line 149 & Page 16, lines 379-381 |

| | |
|---|---|
| | $W_{El}$ depends on the amount of electricity generated by the connected wind farm. This energy is defined as the Annual Energy Production (AEP). It is assumed that the difference between the AEP and $W_{El}$ is fed to the electricity grid.

As shown in Figure 7, $mLCoH_{p,d,P_{El}}$ is obtained at an $r_{El/Farm}$ of about 0.1. A low $r_{El/Farm}$-value results in a high electrolyzer capacity utilization $CF_{El}$, see also Figure 2. Thus, for the use case shown in Figure 5 (a) and discussed in Sect. 3.1, the electrolyzer is therefore almost optimally sized with an $r_{El/Farm}$-value of 0.085, while the rate of 0.17 is above the optimum for use case (b), resulting in LCoH higher than mLCoH. |

| RC | Figure 6: It's hard to tell the two blue lines apart on the figure, especially since they're almost on top of each other. Could one be a different color? |
|---|---|
| AR | Figure 6: We agree that the two blue lines were difficult to distinguish. Therefore, the figure was revised as suggested, using not only different colors for the lines, but also different line styles to ensure that they can be distinguished even when printed in black and white. We also used colors that people who are colorblind can distinguish. |
| AC | Page 15, Figure 6 |
| | - |

| RC | Line 377: Could you describe the scenarios in Figure 9 in a bit more detail up front? |
|---|---|
| AR | The authors agree that the term "scenarios" may be misleading here as there seems to be more to it than a change in fuel price or the exclusion of the pipeline as a distribution mode. Therefore, we have reworded this part to remove the term "scenario" and added the information that the fuel prices used here are exemplary. The fuel price change is used to show qualitatively how it would affect the operating window of the diesel truck. Therefore, the authors feel that no further information is needed on why the fuel price might rise or fall in the future, as there is plenty of research on this. |
| AC | Page 17, lines 411-414 |
| | Figure 9 provides a more detailed analysis of the most favorable distribution mode depending on distance to the POD and daily hydrogen production. Therefore, results are shown for two different exemplary diesel fuel prices and excluding a pipeline as a distribution mode.  |

| RC | Line 394: Does this mean that the hydrogen used for transportation is not taken out of the total hydrogen produced? Can you offer thoughts on how this would affect the dominance of this method over ling distances? |
|---|---|
| AR | No, the hydrogen used for transportation is subtracted from the total amount of hydrogen produced. This adjustment is made by reducing the amount of hydrogen by the truck's hydrogen usage before calculating the final LCoH using Eq. (1). This approach guarantees that all hydrogen system components are sized based on the actual amount of hydrogen produced. This is similar to a hydrogen fuel price that exactly matches the LCoH. To clarify, this is now also stated in the paper. As the amount of hydrogen used there is no cumulative error over long distances. However, a thorough economic analysis of the hybrid farm would be necessary to determine if selling the hydrogen used by the truck would result in higher profits than the cost of diesel fuel for a diesel truck (also considering the higher CAPEX of the hydrogen truck as shown in Table 2). But as the methodology is designed to provide preliminary design for site-specific wind-hydrogen systems, no further deliberation is provided here. |
| AC | Page 18, lines 429-434 |
| | This is because there is no additional cost included in the model for hydrogen consumed by trucks. It is assumed that the hydrogen consumption of the trucks is covered by the production of the wind farm, so that the hydrogen price is equal to the LCoH. This is a simplification, as additional infrastructure is required for hydrogen refueling at the wind farm and the hydrogen used cannot be sold, which must be taken into account in any economic analysis.  |

***Minor Edits:***

AR: All comments have been incorporated, unless they have been made obsolete by rephrasing certain parts.

| RC | Line 42: "folling" -> "following" |
|---|---|
| AC | Page 2, line 60 |
| | … answering the following research … |
| **RC** | **Line 46: niveau is a french word, I believe.  Should be "level" instead?** |
| AC | Page 3, line 66 |
| | What level  of LCoH .. |
| **RC** | **Line 80: "as often used" should be "as is often used"** |
| AC | Page 4, line 102 |
| | be 7 %, as is often used |
| **RC** | **Line 86: "The lifetime is of each..." -> "The lifetime of each ..."** |
| AC | Page 4, line 108 |
| | The lifetime  of each |
| **RC** | **Line 109: "Especially since PEMEL ..." -> "Since PEMEL ..."** |
| AC | Page 5, line 131 |
| |  Since PEMEL have |
| **RC** | **Line 153: "of the electrolyzer. l_cabel." extra period?** |
| AC | Page 7, line 177 |
| | the electrolyzer. $l_{cable}$. includes |
| **RC** | **Line 186: "additional degree of freedom are..." -> "additional degree of freedom is..."** |
| AC | Page 8, line 211 |
| | degree of freedom is  the possible |
| **RC** | **Line 347: "almost linear." -> "almost linearly."** |
| AC | Page 16, line 382 |
| | increase almost linearly. |

| RC | Table 2: in the lifetime column, what does "a" stand for? |
|---|---|
| AR | Referring to the question regarding the a in Table 2: a means annum here and is used throughout the manuscript (e.g. Table 1). For clarification, the unit is now also been introduced as part of the introduction of Eq. (2) and (3). |
| AC | Page 4, line 108 |
| | the parameter $n$, given in years $a$. |